# Calibration techniques for node classification using graph neural networks on medical image data

**I.N. Vos**[1]                                       I.N.VOS-6@UMCUTRECHT.NL
**I.R. Bhat**[1]                                       I.R.BHAT@UMCUTRECHT.NL
**B.K. Velthuis**[2]                                B.K.VELTHUIS@UMCUTRECHT.NL
**Y.M. Ruigrok**[3]                              IJ.M.RUIGROK@UMCUTRECHT.NL
**H.J. Kuijf**[1]                                        H.KUIJF@UMCUTRECHT.NL

[1] *Image Sciences Institute, University Medical Center Utrecht*

[2] *Department of Radiology, University Medical Center Utrecht*

[3] *Department of Neurology and Neurosurgery, University Medical Center Utrecht*

**Editors:** Accepted for publication at MIDL 2023

## Abstract

Miscalibration of deep neural networks (DNNs) can lead to unreliable predictions and hinder their use in clinical decision-making. This miscalibration is often caused by overconfident probability estimates. Calibration techniques such as model ensembles, regularization terms, and post-hoc scaling of the predictions can be employed to improve the calibration performance of DNNs. In contrast to DNNs, graph neural networks (GNNs) tend to exhibit underconfidence. In this study, we investigate the efficacy of calibration techniques developed for DNNs when applied to GNNs trained on medical image data, and compare the calibration performance of binary and multiclass node classification on a benchmark dataset and a medical image dataset. We find that post-hoc methods using Platt scaling or Temperature scaling, or methods that add a regularization term to the loss function during training are most effective to improve calibration. Our results further indicate that these calibration techniques are more effective for multiclass classification tasks compared to binary classification tasks.

**Keywords:** Calibration, graph neural networks, node classification, medical imaging

## 1. Introduction

The trustworthiness of deep neural networks (DNNs) has become an active topic of research over the past years. As DNNs are increasingly being used for real-life decision making, it is important that these models are well-calibrated. This means that, in addition to being accurate for predictions that indicate high confidence, a model should indicate low confidence when it is likely to be inaccurate. This is particularly important in situations where model predictions are utilized for clinical purposes, for instance to aid physicians in making treatment decisions. Studies have shown that most DNNs are prone to miscalibration, often because of an overconfidence in their probability estimates (Guo et al., 2017). Methods to improve the calibration performance include averaging the predictions of multiple models, adding a regularization term during training, or post-hoc scaling of the predictions.

In contrast to DNNs, poor calibration observed in graph neural networks (GNNs) is usually attributed to underconfident estimates (Wang et al., 2021; Liu et al., 2022). Accordingly, calibration improvement techniques should decrease the entropy (thereby increasing the confidence) of probability estimates for correct predictions.

In this work, we investigate if calibration improvement methods developed for DNNs are generalizable to GNNs trained on medical image data. We compare calibration performances for both binary-and multiclass classification using GNNs trained on a benchmark dataset, i.e. the Cora dataset (Sen et al., 2008) and an in-house dataset comprised of 150 3 D time-of-flight Magnetic Resonance Angiography (TOF-MRA) images visualizing the intracranial arterial vasculature (Mensing et al., 2023).

## 2. Materials and methods

### 2.1. Data

#### 2.1.1. Cora dataset

We included the Cora dataset as benchmark, which is a popular citation network dataset and generally considered a benchmark for node classification tasks (Sen et al., 2008). The Cora dataset consists of a single graph with 2,708 nodes and 5,429 edges, divided into 7 classes. Each node has 1,433 features and relates to a paper from a research area or class. We selected the majority class to train a binary classifier, which accounted for 30.2% of the class instances. For multiclass classification, we included all classes.

#### 2.1.2. Medical image dataset

To obtain graphs from medical image data, we used 150 time-of-flight Magnetic Resonance Angiography (TOF-MRA) images from healthy subjects who were scanned on a 3 T Philips MRI system with a voxel size of 0.357 x 0.357 x 1.0 mm. We obtained segmentations of the intracranial arteries using a validated 3 D U-Net (de Vos et al., 2021). A skeletonization method based on a distance transform (Selle et al., 2002) was used to extract the artery skeletons and obtain the radii at different locations. These skeletons were converted into graph representations with nodes relating to artery bifurcations or endpoints and edges relating to connected arteries.

The node classification task focused on predicting the nodes forming the circle of Willis (CoW), an anastomosis of arteries located at the base of the brain. To train a binary classifier, we assigned positive class labels to all nodes or bifurcations forming the CoW. For multiclass classification, we assigned each CoW node a separate class (see Figure 1). All remaining nodes were regarded as background and assigned another class. For each node, the coordinate of its position and its radius were included as features, similarly to (Chen et al., 2020). In addition, we added the number of connected edges and the logarithm of the sum of its connected edge lengths as node feature. The logarithm was used to reduce skewness towards a few relatively large values (e.g. for the internal carotid arteries).

### 2.2. Calibration

We compared node classification results using five methods aimed at improving calibration, which are further described in this section.

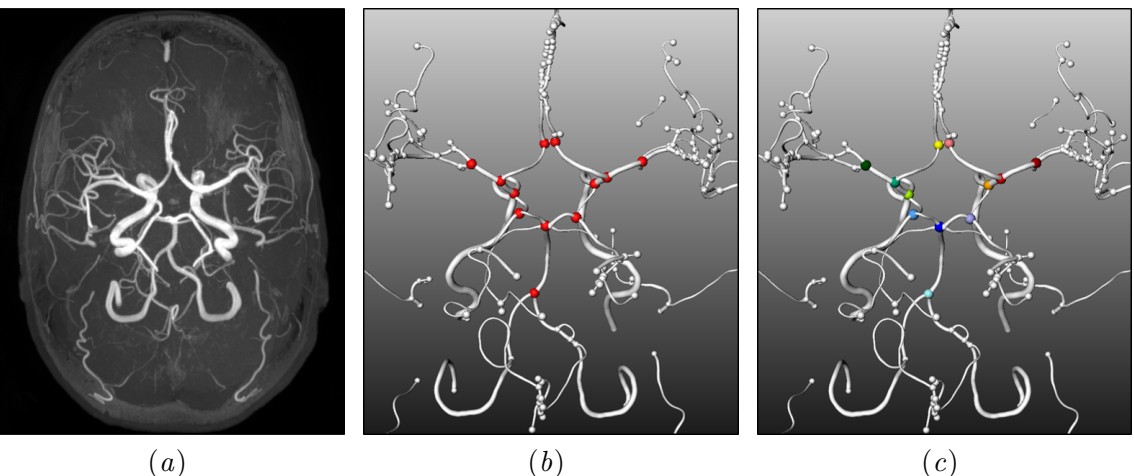

Figure 1: Labelling of nodes. (a) Intracranial arteries on a 3 D TOF-MRA scan. (b) Node labelling for binary classification. (c) Node labelling for multiclass classification. Each color represents a specific target class (background nodes shown in white).

- Platt Scaling (PS): PS involves the fitting of a logistic regression model on the logits (i.e. non-probabilistic outputs) of a neural network (Platt et al., 1999). The logit vector $z$ is used to obtain a calibrated confidence $\hat{q}$ according to:

$$\hat{q}_i = \sigma \left( a z_i + b \right) \tag{1}$$

, where $\sigma$ is the softmax function and $a$ and $b$ are optimized over the validation set.
- Temperature Scaling (TS): TS is a simple extension of Platt scaling (Guo et al., 2017). In contrast to PS, it uses a single scalar parameter $T$ to rescale the logits of a neural network by obtaining $\sigma(z/T)$. Both PS and TS were implemented using the open-source Netcal calibration framework (Küppers et al., 2020).
- Monte-Carlo dropout (MC-dropout): a technique that can be used to approximate Bayesian uncertainty in DNNs. Based on the findings by Gal and Ghahramani (2016), we obtained a distribution of the softmax input values by performing 100 forward passes, using an edge-level dropout layer before the final convolutional layer with a dropout rate of p = 0.5 during inference. Average model outputs were used for further evaluation.
- Model ensemble: we introduced another source of randomization by training multiple models with independent initializations and shuffled training indices. The study by (Fort et al., 2019) used an ensemble of 3 independent models to demonstrate that randomly initialized neural networks converge to different local optimal, which can benefit the performance in terms of uncertainty and accuracy. Given the low computational cost of graphs, we trained 10 independently initialized models and used the average model outputs to evaluate its performance.
- Accuracy-versus-uncertainty (AvU): we evaluated the use of a regularization term adapted from the study by Mody et al. (2022). The proposed loss function captures

the relation between accuracy and uncertainty, and reaches a minimum when all accurate voxels (here, nodes) are certain and all inaccurate voxels are uncertain. We follow their approach and compute the average AvU loss for a range of certainty thresholds:

$$loss_{AvU} = \frac{1}{T} \sum_{t=1}^{T} ln \left(1 + \frac{n_{au}^t + n_{ic}^t}{n_{ac}^t + n_{iu}^t}\right) \tag{2}$$

, where T is the number of thresholds and $n$ the number of nodes that are accurate ($a$) or inaccurate ($i$) and certain ($c$) or uncertain ($u$). The final loss function is the sum of the cross entropy loss and $\alpha$*AvU loss, with $\alpha$ optimized to 0.1 for our experiments.

- Graph calibration loss (GCL): the study by Wang et al. (2022) addressed the under-confidence of GNNs by adding a minimal-entropy regularization item to the cross-entropy loss function (referred to as the calibration hyperparameter $\gamma$). We optimized this parameter to $\gamma = 1.5$ and implemented the GCL loss as follows:

$$loss_{GCL} = -\sum_{y=1}^{K} (1 + \gamma \hat{p}_y) \, p_y \, log \, \hat{p}_y \tag{3}$$

, where $p$ is the target distribution and $\hat{p}$ is the predicted distribution.

### 2.3. Experiments

All experiments were performed using PyTorch Geometric (version 1.12.1). We evaluated the classification performance using balanced accuracy (BA) to account for class imbalance. Miscalibration was assessed with reliability diagrams (Niculescu-Mizil and Caruana, 2005) and the expected calibration error (ECE) (Naeini et al., 2015). A reliability diagram uses a partitioning scheme to plot the mean confidence score against the fraction of positive class samples, or prevalence, per bin. Perfect calibration is obtained when the diagram is plotted as a diagonal line. The ECE quantitatively measures miscalibration as the expected absolute difference between confidence and accuracy, ranging between [0,1]. Similar to reliability diagrams, confidence scores are partitioned into $M$ equally-spaced bins. The ECE is approximated by taking the weighted average of the bin's absolute difference between confidence and accuracy. We used the Netcal framework (Küppers et al., 2020) to perform histogram binning with $M = 10$ and obtain the reliability diagrams and ECE scores.

#### 2.3.1. Cora dataset

The dataset was divided into training (140 nodes), validation (500 nodes) and test (1000 nodes) masks based on the public fixed split reported in Yang et al. (2016). We trained a two-layer graph convolutional network (GCN) (Kipf and Welling, 2016) using a hidden layer dimension of 16. The learning rate was set to 0.01, the weight-decay to 5e-3 and the number of epochs to 200. The Adam optimizer was used for parameter optimization. Training was stopped if the validation loss had not decreased for 10 successive iterations. Code to reproduce results is available on github (https://github.com/irisnadinevos/cora-calibration-techniques).

### 2.3.2. MEDICAL IMAGE DATASET

The graphs from the medical image dataset were split into a training (N=80), validation (N=20), and test set (N=50). A two-layer GCN was trained with a hidden layer dimension of 16, a dropout rate of 0.1, and a batch size of 25. We used a learning rate of 0.01, weight decay of 5e-4, and the Adam optimizer. Early stopping was applied with a window size of 20. Because of the large amount of background nodes, with only 4.97% of the nodes constituting to target classes, the ECE was evaluated for smaller, local regions in addition to the graph on a global-level (i.e. including all nodes). These local regions were defined by including a maximum of three neighbouring nodes for each target node (see example in Figure 4, Appendix A). The BA and ECE were calculated on a subject-level and tested for statistically significant differences between methods using a Wilcoxon signed-rank test. In addition to the vanilla GCN, we trained a second two-layer GCN using higher-order graph convolutions as proposed in Morris et al. (2019). Parameter settings were kept similar to the ones described in this section.

## 3. Results

Reliability diagrams are shown in Figure 2 for the Cora dataset and in Figure 3 for the medical image dataset. Table 1 and 2 summarize the results in terms of BA and ECE per method. For all uncalibrated models, we observed an underconfidence. ECE scores for multiclass classification were lower than for binary classification, for both datasets.

### 3.1. Cora dataset

Binary node classification on the Cora dataset yielded a BA of 0.76 and an ECE of 0.124 (uncalibrated). For multiclass node classification, a BA of 0.83 and an ECE of 0.089 (uncalibrated) were obtained. Both binary and multiclass classification showed best calibration was obtained using post-hoc PS (ECE 0.040 and 0.025) and TS (ECE 0.113 and 0.029), where scores for BA using PS increased for binary (BA 0.86) and slightly decreased for multiclass (BA 0.80) classification. No improvements were observed using MC-dropout or model ensembles. For binary classification, the AvU loss resulted in better calibrated models (ECE 0.100) with higher accuracy (BA 0.80), wheares using the GCL showed no improvements. For multiclass classification, both the AvU loss (ECE 0.033) and GCL (ECE 0.034) yielded improvements in calibration.

The reliability diagram for binary classification shows that the confidences calibrated with the PS method are closed to the diagonal line. TS and the AvU loss also resulted in better calibrated confidences compared to the other methods. For multiclass classification, the diagrams of both post-hoc methods (PS and TS) and regularization losses (AvU and GCL) converge for confidences > 0.6.

### 3.2. Medical image dataset

The mean number of nodes per graph was $171 \pm 30$. The local graph regions (based on a maximum of three neighboring nodes) included a mean number of $29 \pm 6$ nodes.

A mean BA of 0.58, mean local ECE of 0.236 and mean global ECE of 0.030 was achieved for uncalibrated, binary node classification on the MI dataset. ECE scores after TS

showed a calibration improvement only locally. For MC-dropout and the model ensemble, no improvements in calibration were observed. Notably, the AvU loss and GCL resulted in worse discrimination and calibration compared to the uncalibrated model. Similar to the Cora dataset, the reliability diagram shows that the results for PS are closest to the diagonal line. The GCL method reveals higher underconfidence compared to the uncalibrated model.

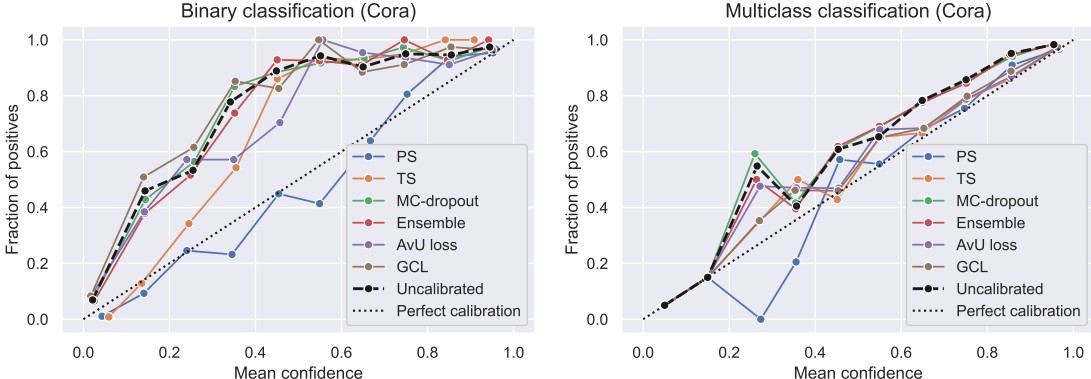

Figure 2: Reliability diagrams for different calibration techniques on the Cora dataset. Left: binary node classification. Right: multiclass node classification.

Table 1: Results on the Cora dataset. BA = balanced accuracy; ECE = expected calibration error; PS = Platt scaling; TS = Temperature scaling; MC-dropout = Monte-Carlo dropout; AvU = accuracy-versus-uncertainty; GCL = graph calibration loss. Best calibration performances are highlighted in bold.

| CORA | Binary classification | | Multiclass classification | |
|---|---|---|---|---|
| | BA | ECE | BA | ECE |
| Uncalibrated | 0.76 | 0.124 | 0.83 | 0.089 |
| PS | **0.86** | **0.040** | **0.80** | **0.025** |
| TS | 0.76 | 0.113 | 0.83 | 0.029 |
| MC-dropout | 0.76 | 0.124 | 0.83 | 0.089 |
| Ensemble | 0.76 | 0.123 | 0.83 | 0.089 |
| AvU loss | 0.80 | 0.100 | 0.82 | 0.033 |
| GCL | 0.77 | 0.128 | 0.83 | 0.034 |

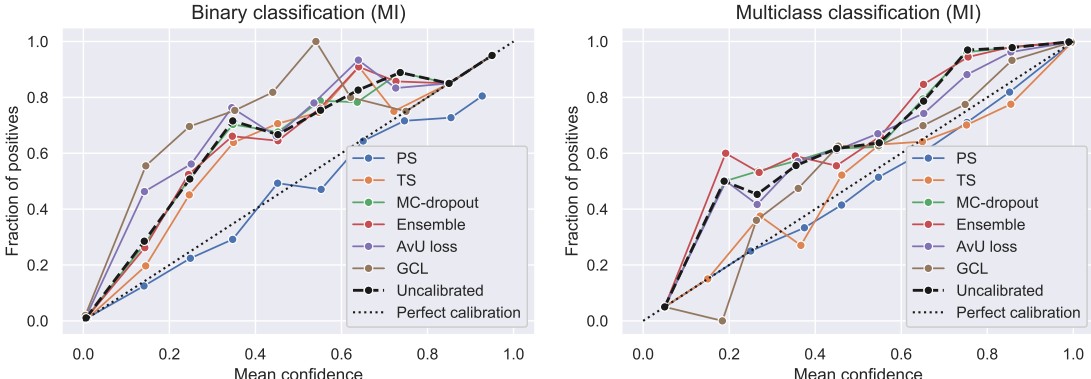

Figure 3: Reliability diagrams for different calibration techniques on the medical image (MI) dataset. Left: binary node classification. Right: multiclass node classification.

Table 2: Results on the medical image (MI) dataset. BA = balanced accuracy; ECE = expected calibration error; PS = Platt scaling; TS = Temperature scaling; MC-dropout = Monte-Carlo dropout; AvU = accuracy-versus-uncertainty; GCL = graph calibration loss. * means statistical significant difference (p<0.05) when compared to uncalibrated models, bold values denote significant improvements.

| MI | Binary classification | | |
|---|---|---|---|
| | BA | ECE local ($\times 10^{-2}$) | ECE global ($\times 10^{-2}$) |
| Uncalibrated | 0.58 $\pm$ 0.07 | 23.6 $\pm$ 4.8 | 3.0 $\pm$ 1.0 |
| PS | 0.58 $\pm$ 0.07 | **19.5 $\pm$ 5.5**$^*$ | **2.9 $\pm$ 1.1**$^*$ |
| TS | 0.58 $\pm$ 0.07 | **21.9 $\pm$ 4.9**$^*$ | **2.8 $\pm$ 0.9**$^*$ |
| MC-dropout | 0.58 $\pm$ 0.08 | 23.1 $\pm$ 5.4 | 3.0 $\pm$ 1.0 |
| Ensemble | 0.58 $\pm$ 0.08 | **22.8 $\pm$ 5.1**$^*$ | 3.0 $\pm$ 1.0 |
| AvU loss | 0.56 $\pm$ 0.06$^*$ | 25.1 $\pm$ 4.5$^*$ | 3.3 $\pm$ 1.1$^*$ |
| GCL | 0.51 $\pm$ 0.03$^*$ | 27.8 $\pm$ 4.5$^*$ | 3.7 $\pm$ 1.1$^*$ |
| | Multiclass classification | | |
| | BA | ECE local ($\times 10^{-2}$) | ECE global ($\times 10^{-2}$) |
| Uncalibrated | 0.69 $\pm$ 0.19 | 23.4 $\pm$ 6.3 | 4.9 $\pm$ 1.4 |
| PS | 0.69 $\pm$ 0.19 | **19.0 $\pm$ 5.3**$^*$ | **2.6 $\pm$ 1.1**$^*$ |
| TS | 0.69 $\pm$ 0.19 | **19.0 $\pm$ 6.3**$^*$ | **3.0 $\pm$ 1.3**$^*$ |
| MC-dropout | 0.67 $\pm$ 0.17$^*$ | 23.6 $\pm$ 5.5 | 4.9 $\pm$ 1.2 |
| Ensemble | 0.65 $\pm$ 0.18$^*$ | 23.8 $\pm$ 5.9 | 5.0 $\pm$ 1.4 |
| AvU loss | 0.67 $\pm$ 0.19 | 22.7 $\pm$ 6.1 | **3.9 $\pm$ 1.1**$^*$ |
| GCL | 0.64 $\pm$ 0.19$^*$ | **20.0 $\pm$ 6.0**$^*$ | **3.2 $\pm$ 1.2**$^*$ |

For multiclass classification, we obtained a mean BA of 0.69, mean local ECE score of 0.234 and mean global ECE score of 0.049. In line with the previous results, TS improved calibration performances on a local- and global-level. MC-dropout showed small improvements in ECE scores at the cost of accuracy. The calibration performance of the model ensemble did not improve compared to the uncalibrated model. Both the AvU loss and GCL resulted in better calibrated models. The reliability diagram indicates best performances were obtained for PS, TS, and the GCL loss.

The results of node classification using higher-order graph convolutions are shown in Table 3 (Appendix B).

## 4. Discussion and conclusion

In this work, we compared the performance of several calibration techniques for binary and multiclass node classification on the Cora dataset and a medical image dataset. Our results showed that multiclass node classification yielded better calibrated and more discriminative models than binary classification. Overall, we found that best calibration performances were obtained using post-hoc Platt scaling (PS), Temperature scaling (TS), or by adding a regularization term during training (AvU loss or GCL).

In most cases, PS outperformed TS. This difference can be mathematically explained by the variable $b$ (bias) in Equation 1, which is not included in TS. Indeed, similar results to TS were observed for binary classification on the Cora dataset when we removed the bias (see Figure 5 in Appendix C). The learned parameters for TS, which are shown in Table 4 in Appendix D, indicate that temperature (T) $< 1$ for binary classification and T $> 1$ for multiclass classification. In general, TS with T $< 1$ will decrease the entropy of the probability distribution, whereas T $> 1$ will increase the entropy. From the reliability diagrams we can observe that for values of T closer to 1, the calibrated confidences more closely resemble uncalibrated confidences.

Calibration techniques such as MC-dropout and a model ensemble, which are based on average predictions of multiple models, did not generally improve calibration performances on both datasets. Although previous studies have shown that ensembles trained with random initializations perform well (Fort et al., 2019), the underperformance observed in this study is likely attributed to the two-layer GCN architectures and their lack of diversity.

The AvU loss and GCL yielded worse performances than the uncalibrated model for binary classification on the medical image dataset. This may be caused by a lack of model capacity, as BA scores were relatively low. The performance for Cora is possibly affected by binarization of the node labels. As we kept the original number of samples per class in the training set, there is a higher class imbalance than in the multiclass training set. Interestingly, the BA scores, reliability diagrams and ECE scores for binary classification indicate that the AvU loss is superior to GCL on both datasets. By explicitly defining the accuracy and uncertainty in the loss function, the AvU loss seems to be more robust to class imbalance. In Mody et al. (2022), the authors propose the use of an additional loss term that penalizes high probability of uncertainty in large accurate regions (e.g. at the core of a segmented organ). Such implementations or further optimization of the loss function may also improve the current calibration performance.

We found lower mean ECE scores on a global-level than a local-level. The study of Hsu et al. (2022) discussed several factors affecting GNN calibration, among which the neighbourhood similarity of nodes, and observed nodes that have similar label predictions as their neighbours tend to have lower calibration errors. This potentially explains the higher ECE scores of local nodes: the nodes surrounding and representing the CoW have different label predictions and may be inclined to have similar predictions as their neighbours.

To compare the results on the Cora dataset to other studies, we implemented the most commonly used graph convolution (Kipf and Welling, 2016). Although the model calibration was relatively good, the discriminative power on the medical image dataset was low. The results in Table 3 show that other graph convolutional operators, such as the one proposed by Morris et al. (2019), can yield better calibration and discrimination.

The BA obtained for multiclass classification on the Cora dataset was comparable to the accuracy scores reported in literature (range 0.814-0.816) (Wang et al., 2021; Liu et al., 2022). Our ECE scores for the uncalibrated model and TS model were slightly lower than the scores reported in Wang et al. (2022). This may be caused by minor differences in model architecture (e.g. no dropout) or feature normalization. We did not perform node feature normalization because this did not prove to be beneficial on the MI dataset, where we included the node coordinates as features. Regardless, we obtained a similar ECE score on the Cora dataset using GCL during training.

Our results show that calibration techniques developed for DNNs to mitigate overconfidence can improve calibration for underconfident GCN models trained on medical image data. Post-hoc scaling using PS or TS, or using a regularization term during training showed overall improvements in calibration performance. Although these findings have implications for future studies using GCNs on medical image data, more research on the calibration of GCNs is needed in order to use these models for clinical decision-making.

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

## Appendix A. Expected calibration error (ECE)

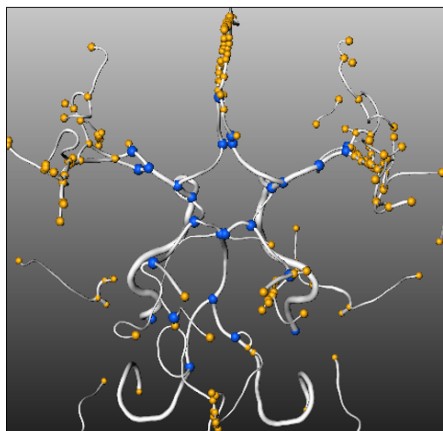

Figure 4: Visualization of nodes considered for calculation of the expected calibration error (ECE) on a sample from the medical image dataset. Nodes marked with a blue color were used to calculate local ECE scores. Nodes marked with an orange color represent the remaining nodes. For global ECE scores, all nodes were included.

## Appendix B. Higher-order graph convolution

Table 3: Results on the medical image (MI) dataset using the graph convolutional operator proposed by Morris et al. (2019). BA = balanced accuracy; ECE = expected calibration error; PS = Platt scaling; TS = Temperature scaling; MC-dropout = Monte-Carlo dropout; AvU = accuracy-versus-uncertainty; GCL = graph calibration loss. The mean and standard deviation are calculated across all subjects. * represents a statistical significant difference (p<0.05) when compared to uncalibrated models, bold values denote significant improvements.

| MI | Binary classification | | |
|---|---|---|---|
| | BA | ECE local (x10$^{-2}$) | ECE global (x10$^{-2}$) |
| Uncalibrated | 0.73 ± 0.11 | 19.4 ± 4.7 | 2.7 ± 1.0 |
| PS | 0.73 ± 0.11 | **15.8 ± 4.5**$^*$ | **2.4 ± 1.0**$^*$ |
| TS | 0.73 ± 0.11 | 19.4 ± 4.5 | 2.6 ± 0.8 |
| MC-dropout | 0.73 ± 0.11 | 19.3 ± 4.7 | 2.7 ± 0.9 |
| Ensemble | **0.77 ± 0.10**$^*$ | **18.0 ± 4.4**$^*$ | 2.6 ± 0.9 |
| AvU loss | **0.78 ± 0.10**$^*$ | **17.7 ± 5.3**$^*$ | **2.5 ± 1.1**$^*$ |
| GCL | 0.75 ± 0.10 | 18.9 ± 5.5 | 2.7 ± 1.1 |
| | Multiclass classification | | |
| | BA | ECE local (x10$^{-2}$) | ECE global (x10$^{-2}$) |
| Uncalibrated | 0.80 ± 0.14 | 16.8 ± 4.5 | 2.7 ± 0.8 |
| PS | 0.80 ± 0.14 | **14.2 ± 6.1**$^*$ | **2.1 ± 1.1**$^*$ |
| TS | 0.80 ± 0.14 | **14.4 ± 5.8**$^*$ | **2.3 ± 1.0**$^*$ |
| MC-dropout | 0.80 ± 0.14 | 16.3 ± 4.7 | 2.7 ± 0.8 |
| Ensemble | **0.87 ± 0.14**$^*$ | 17.7 ± 4.7 | 2.9 ± 0.8 |
| AvU loss | **0.85 ± 0.13**$^*$ | **15.2 ± 4.9**$^*$ | **2.3 ± 0.8**$^*$ |
| GCL | **0.88 ± 0.12**$^*$ | **12.9 ± 5.1**$^*$ | **2.0 ± 0.8**$^*$ |

## Appendix C. Platt scaling (PS)

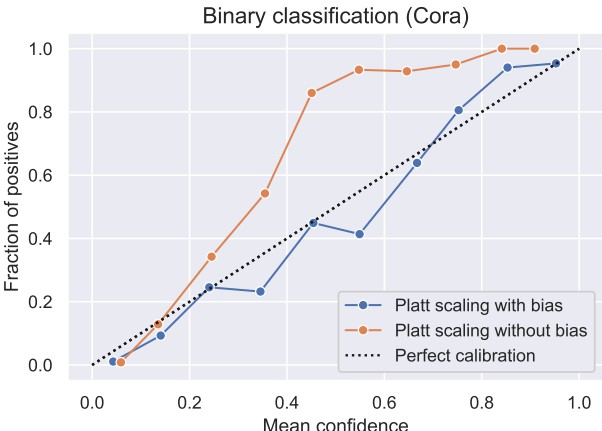

Figure 5: Reliability diagram of confidences obtained from binary node classification on the Cora dataset using Platt scaling with and without bias.

## Appendix D. Temperature scaling (TS)

Table 4: Learned temperature (T) parameters for binary and multiclass classification on the Cora dataset and the medical image (MI) dataset.

|  | Binary classificaion | Multiclass classification |
| --- | --- | --- |
| Cora dataset | 0.529 | 1.241 |
| MI dataset | 0.837 | 1.870 |

