# OpenReview forum: "Calibration techniques for node classification using graph neural networks on medical image data"
_MIDL.io/2023/Conference — MIDL 2023 Poster_

### Official Review · Reviewer_F3wm · 2023-02-03

**Confidence:** 4
**Preliminary Rating:** 3
**Recommendation:** Poster

**Summary:**

The authors apply and evaluate different calibration techniques used in deep neural networks in graph neural networks. The problems used to illustrate the different techniques are two: a database of publications where the task is to infer the research area of a publication, and a medical imaging database where a graph is constructed from the intercranial arteries segmented from MRI data, and the task is to detect the nodes related to the circle of Willis (a circle within the arterial network).

**Strengths:**

Calibration of neural networks is often overlooked. Probabilities are more often than not taken as measurements of goodness, forgetting that they should represent the proportion of specimens that are truly positive for that class. This works puts its focus into such a valid problem, providing data on the performance of several modern calibration techniques.

**Weaknesses:**

The paper could explain better the problem of calibration, as it surely will come as a surprise to many researchers. Also, better explanation of the expected calibration error would be beneficial for the reader. How many bins were used?
It is hard to understand that the multiclass problem is better calibrated than the binary classification.
The paper is purely observational, an evaluation of existing methods to two problems.



**Deanonymize Review:**

no

**Paper Type:**

validation/application paper

**Questions To Address In The Rebuttal:**

This reviewer wonders if other simpler calibration methods, such as logistic regression on the output of the network, would work as well as the methods evaluated. As the authors describe, it is a simple scaling of the predictions.

---

### Official Review · Reviewer_QJPX · 2023-02-04

**Confidence:** 4
**Preliminary Rating:** 4

**Summary:**

The paper discusses techniques of calibration when working with Graph Neural Networks (GNN). More specifically, the authors study several methods used for deep neural networks, such as Temperature Scaling, MC-Dropout, Loss regularization, and others. The authors run experiments on two datasets: Graph dataset Cora, and a private dataset generated from Magnetic Resonance Angiography. The paper is concluded with Temperature Scaling being the best-performing method in the studied scenarios.

**Strengths:**

The authors study a relevant issue of calibration of algorithms to allow for more reliable predictions. They focus on graph neural networks which are less common than more traditional Deep Convolutional Neural Networks and hence could raise the interest of the audience.

**Weaknesses:**

While the paper focuses on an interesting and relevant topic, it seems to lack details to allow for a clear understanding and an appropriate conclusion. That is, the evaluated methods are introduced briefly without detailed descriptions (e.g., equations); multiple metrics are reported without an in-depth discussion of their meaning and relevance. Hence, it may be hard for the reader to understand the conclusion and the overall take-home message. Therefore, details and clarifications are needed for a better understanding of the paper.

**Deanonymize Review:**

yes

**Detailed Comments:**

- In the introduction, could the authors clarify/rephrase the definition of "true underlying probabilities"?
- In the introduction, could the authors provide references to the datasets used
- In 2.1.2 could the authors provide more details about how the features were selected?
- In 2.2, could the authors indicate, what the \lambda value corresponds to? How the N was selected?
- In 2.2 (AvU) Could the authors indicate when \alpha is 0.1 and when it is 1.0?
- At the beginning of 2.3, could the authors provide more details on the ECE metric as it might not be well-known by all readers?
- In 2.3.1, could authors detail the motivation for selecting only 140 nodes for training (15% of the test)?
- The 3.1 seems to have repeating content, e.g. the fact that "MC-dropout and the model ensemble did not affect calibration performances" is mentioned twice. Could the authors revise it? Moreover, the ECE and Accuracy of the TS method are not reported.
- In Results, could the authors discuss more the reliability diagrams? Also, could they discuss the ECE values more in-depth?
- Could the authors revise Table 1 to improve readability? In particular, the top results are not highlighted, the ECE values are reported as x10^-2, so it is not simple to refer to the values from the text.
- In the Discussion the authors state the TS as the top-performing method. Could they discuss more the scaling parameters used?  Also, could the authors put more light on the addition of the regularization term as the tables (Table 2) do not allow for such a conclusion.
- In the Discussion the authors mention being limited by the GNN architecture. Could they state if any additional experiments were performed to verify this claim?

**Paper Type:**

methodological development

**Questions To Address In The Rebuttal:**

I would expect the following to be addressed in the rebuttal:
- Could the authors provide more details on the methods being evaluated, in particular, provide equations for each of the tested methods?
- Could the authors revise the discussion of the results (taking into account the detailed comments) to allow for a better understanding of the advantages and short-comings of the methods tested?
- Could the authors provide more details about the tested methods, in particular the choice of the hyper-parameters?
- Could the authors test other GNN architectures to support the statements in the Discussion section?

---

### Official Review · Reviewer_eb2L · 2023-02-06

**Confidence:** 4
**Preliminary Rating:** 4
**Recommendation:** Poster

**Summary:**

This is a clear and well-written paper with a clear goal: test various calibration techniques known for deep convolutional neural networks for graph neural networks. Several well-known and often applied calibration techniques are tested and the results demonstrate that temperature scaling gave the best results for graph neural networks. This is a clear and useful paper for the community, and present clear advice for future work with graph neural networks.

**Strengths:**

- Clear goal and well-written
- Good description of the various approaches that are used for calibration
- Clear introduction
- Comprehensive discussion of the observed results and directions for future work

**Weaknesses:**

- In general, there are few weakness to this paper.
- It would help to add a public github repo for the experiments with the Cora dataset so that these resutls can be reproduced by other authors
- The presented medical data set is not publicly available

**Deanonymize Review:**

no

**Detailed Comments:**

It would have liked to see Platt's scaling as another calibration method in this paper.

**Paper Type:**

methodological development

**Questions To Address In The Rebuttal:**

- Please add a public github repo for the experiments with the Cora dataset so that these results can be reproduced by the community.
- Would it be possible to add Platt's scaling as another calibration method?

---

### Meta-Review · Area_Chair_1zqy · 2023-02-25

**Recommendation:** Accept (Poster)
**Confidence:** 3

**Metareview:**

The authors present a study comparing multiple calibration techniques for graph neural networks (GNNs). Experiments were performed on 2 datasets, the Cora publications dataset and a private medical imaging dataset. They found that GNNs tend to show underconfidence, unlike other DNNs.

Reviewers agreed on the interest of the topic of calibration, which is often overlooked, and the clarity of the goals of the paper. Multiple concerns regarding an important missed comparison method (Platt scaling) was added by the authors in revision. Concerns regarding a number of methodological details were addressed by authors in the rebuttal. There were also concerns regarding evaluation on a private medical dataset and the "purely observational" nature of the work.

While I agree calibration is an important problem that often does not get addressed and the addition of Platt scaling comparison greatly improves the paper, I also agree with concerns on choice of dataset for evaluation. In addition to the use of the private dataset, the authors test on the Cora dataset as a benchmark, which is not at all specific to medical imaging (it is a publications dataset covering general machine learning papers). Given the many public medical imaging datasets that could have been chosen as baseline and the goal of comparing calibration techniques for GNNs "on medical image data", this choice of dataset is a weakness. Further, while an observational study can certainly generate interesting knowledge and could provide a nice tutorial on calibration techniques, the limited evaluation on 1 public and 1 relatively smaller private dataset weakens what are meant to be general conclusions drawn by the authors.

Still, the revised paper is much improved and the paper topic would be of strong interest to the MIDL community, so I recommend accept.